# Autism and Migraine: An Unexplored Association?

**DOI:** 10.3390/brainsci10090615

**Published:** 2020-09-06

**Authors:** Luigi Vetri

**Affiliations:** 1Department of Sciences for Health Promotion and Mother and Child Care “G. D’Alessandro”, University of Palermo, 90127 Palermo, Italy; luigi.vetri@gmail.com or luigi.vetri@you.unipa.it; Tel.: +39-3286434126; 2Oasi Research Institute-IRCCS, 94018 Troina, Italy

**Keywords:** autism, ASD, migraine, headache, pain

## Abstract

Autism spectrum disorder is characterized by neurological, psychiatric and medical comorbidities—some conditions co-occur so frequently that comorbidity in autism is the rule rather than the exception. The most common autism co-occurring conditions are intellectual disability, language disorders, attention-deficit hyperactivity disorder, epilepsy, gastrointestinal problems, sleep disorders, anxiety, depression, obsessive-compulsive disorder, psychotic disorders, oppositional defiant disorder, and eating disorders. They are well known and studied. Migraine is the most common brain disease in the world, but surprisingly only a few studies investigate the comorbidity between autism and migraine. The aim of this narrative review is to explore the literature reports about the comorbidity between autism and migraine and to investigate the common neurotransmitter, immune, anatomical and genetic abnormalities at the base of these two conditions.

“I had noticed both ‘migraine’ and ‘autism’ listed on the top of my son’s forms and I began to suspect that all of this paperwork had far more to do with the neurologist’s interests in my son’s autism than with a genuine concern for his migraine problems”.Anderson, J. L. (2013). A dash of autism. The philosophy of autism.

## 1. Introduction

Autism spectrum disorder (ASD) is a complex neurobehavioral and neurodevelopmental condition characterized by difficulties in social interaction and communication, restricted and repetitive patterns of behavior, interests and activities and altered sensory processing (see panel 1) [1]. 

The prevalence of autism has significantly increased during the last two decades from 2–5/10,000 to 1/59 children (1 in 37 boys and 1 in 151 girls), the frequency in males is four times greater than females [2].

According to a dimensional view of ASD, it is a heterogeneous neurodevelopmental condition characterized by a spectrum of behaviors and traits and by a complex etiopathogenesis [3]. In addition, to complicate the complex picture of autism, there are a lot of co-occurring conditions, some of these are so frequent that it is difficult to understand if they are a comorbidity or a coexisting trait. Most common ASD comorbidities are: neurodevelopmental disorders (intellectual disability, language disorders, attention-deficit hyperactivity disorder, tic disorders, motor abnormality); general medical conditions (epilepsy, gastrointestinal problems, immune dysregulation, genetic syndromes, sleep disorders); psychiatric disorders (anxiety, depression, obsessive-compulsive disorder, psychotic disorders, substance use disorders, oppositional defiant disorder, eating disorders, personality disorders) [4,5].

Patients with ASD who have such a multidimensional impairment often have greater behavioral abnormalities and worse long-term outcomes [6]. Some authors refer to this condition as “autism-plus” or “multiple complex developmental disorder” [7,8].

Migraine is the most predominant primary headache and represents the most prevalent neurological disease, the third most prevalent illness in the world, the third cause of disability in under 50s and it affects about 1 billion people worldwide [9,10].

According to the International Classification of Headache Disorders, 3rd edition beta (ICHD-3 beta), people suffering from migraine experience attacks lasting 4–72 h which are typically unilateral, pulsating, with a moderate or severe intensity, aggravated by physical activity and associated with nausea, vomiting and/or photophobia and phonophobia. In some patients, head pain is preceded by visual, sensory or other central nervous system symptoms, this presentation is known as aura (see panel 2) [11].

Both people with autism and people with migraine share an atypical sensory processing. Hyper- and hyposensory reactivity can combine differently across individuals with ASD, it can range from mild to severe forms, it can be visible early in development and persist in adulthood. Sensory processing abnormalities seem to have a double impairment both in the registration and modulation of sensory stimuli [12]. 

Sensory abnormalities reflect neurochemical and neuroanatomical alterations, among them GABAergic signaling is often frequently affected [13]. Several studies have detected different structural abnormalities such as volume changes in primary sensory regions, the thalamus as a relay station for most senses, anterior cingulate cortex, insula, amygdala, hippocampus, and cerebellum [14]. 

Similarly, individuals with migraine have an impaired perception and processing of unimodal and multimodal sensory inputs, both during and after migraine attacks [15]. People with migraine during and between migraine attacks have atypical stimulus induced activations of the brainstem and of cortical and subcortical regions that participate in sensory processing. For instance, a resting state functional connectivity MRI study evidenced a stronger connectivity between the periaqueductal gray and thalamus, insula, supramarginal, precentral, and postcentral gyri in individuals with migraine [16]. Migraineurs also show an increased cortical hyperexcitability both during and between migraine attacks [17]. 

Sensory processing impairment could have a genetic component because in siblings of affected people dysfunctional sensory traits have been found to a greater extent than the general population [18].

Although ASD and migraine are very common neurological conditions and despite common findings about their sensory processing impairment, there is a lack of evidence about their co-occurrence [19,20] and only a few works in the scientific literature face this apparently unusual comorbidity.

**Panel 1. DSM-5 Autism Diagnostic Criteria** [1]
Persistent deficits in social communication and social interaction across multiple contexts, as manifested by the following, currently or by history.Restricted, repetitive patterns of behavior, interests, or activities, as manifested by at least two of the following, currently or by history.Symptoms must be present in the early developmental period (but may not become fully manifest until social demands exceed limited capacities or may be masked by learned strategies in later life).Symptoms cause clinically significant impairment in social, occupational, or other important areas of current functioning.These disturbances are not better explained by intellectual disability (intellectual developmental disorder) or global developmental delay. Intellectual disability and autism spectrum disorder frequently co-occur; to make comorbid diagnoses of autism spectrum disorder and intellectual disability, social communication should be below that expected for general developmental level.

**Panel 2. ICHD-3 beta migraine criteria** [11]


**Migraine without aura**
At least five attacks fulfilling criteria B–D.Headache attacks lasting 4–72 h (untreated or unsuccessfully treated).Headache has at least two of the following four characteristics:unilateral location;pulsating quality;moderate or severe pain intensity;aggravation by or causing avoidance of routine physical activity (e.g., walking or climbing stairs).During headache at least one of the following:nausea and/or vomiting;photophobia and phonophobia.Not better accounted for by another ICHD-3 diagnosis.



**Migraine with aura**
At least two attacks fulfilling criteria B and C.One or more of the following fully reversible aura symptoms:visual;sensory;speech and/or language;motor;brainstem;retinal.At least three of the following six characteristics:at least one aura symptom spreads gradually over ≥5 min;two or more aura symptoms occur in succession;each individual aura symptom lasts 5–60 min;at least one aura symptom is unilateral;at least one aura symptom is positive;the aura is accompanied, or followed within 60 min, by headache.Not better accounted for by another ICHD-3 diagnosis.


## 2. Aims and Methods 

This narrative review evaluates the scientific literature evidence about the co-occurrence between ASD and migraine in the hope of shedding some light on this poorly explored association. 

We want to better delineate the main state-of-the-art research findings about the comorbidity between ASD and migraine suggesting the possible related pathophysiological mechanisms and evaluating if patients with ASD are vulnerable to under-recognition and undertreatment of migraine.

To this end, several articles published over the years were reviewed by performing a search using the following syntax “autism” (Title/Abstract) OR “Asperger” (Title/Abstract) OR “pervasive developmental disorders” (Title/Abstract) AND “migraine” (Title/Abstract). References were identified through electronic database searching in CENTRAL, Ovid MEDLINE, Embase, PsycINFO. 

The final database search was run on July 2020.

## 3. Co-Occurring ASD-Migraine Epidemiology

Many physicians who care for people with ASD believe that headache is a marginal uncommon issue rather than a very prominent problem. There are, in the literature, only a few studies that aim to grasp the scale of the phenomenon by analyzing the comorbidity between ASD and migraine (Table 1). However, despite the limited number of studies and their restricted samples, they seem to describe a different scenario.

Underwood et al. in their cohort study of 2019 aimed to examine the phenotypic and genetic characteristics of a sample of adults with ASD and their comorbidities—they found a higher rate of psychiatric comorbidities (89.5% of cases) than in controls. The most frequent comorbid diagnoses were depression (62.9%) and anxiety (55.2%). Moreover, authors demonstrated an increased reported rate of migraine: 42.7% of individuals with ASD reported a lifetime history of migraine compared to 20.5% of controls (*p* = 0.012). There was also a higher rate of epilepsy and seizures compared to controls and an association between migraine and epilepsy was confirmed (*p* = 0.028) [21].

In 2014, Victorio in his study on a small sample of eighteen patients with autism found migraine in 61% (11/18) of patients. Eight patients had migraine without aura, one had migraine with aura and two patients had both migraine with and without aura, three patients had combined migraine and tension type headache and three had chronic daily headache. The onset age widely varies between 5–16 years [22].

In 2014, Sullivan et al. in their sample of patients with autism evidenced migrainous symptomatology in 28.4% of cases. There were no differences in gender, in age or in autism severity between migraineurs and nonmigraineurs. However, children with ASD and migraines significantly showed more generalized anxiety and sensory hyperreactivity suggesting a possible subtype of ASD [23].

## 4. Sensitivity to Pain in ASD

Some authors suggest that a possible explanation of the unusual association between migraine and ASD is to be sought in the hyposensitivity to pain of individuals with ASD [24,25,26].

Sensory anomalies are a hallmark of autism. A global hypersensitivity or hyposensitivity are the most common sensory anomalies, but sometimes some individuals can be hypersensitive in some sensory modalities and hyposensitive in others [27]. In the context of pain sensitivity, another element that should not be underestimated is the difficulty and the different modality of individuals with ASD to communicate pain sensation.

A literature review by David J. Moore highlighted some interesting data about sensitivity to pain in ASD. The evidence about insensitivity to pain, for example, is based on clinical accounts or self/parent report modalities full of interpretations and selective reporting biases [28].

A study with proper systematic examinations and protocols to evaluate pain thresholds showed that ASD group had reduced pain thresholds compared to the control group. Therefore, rather than insensitivity to pain, ASD individuals seem to be hypersensitive to pain and insensitivity could represent a peculiar characteristic of a neurobiological subtype of ASD [29]. Moreover, a reduced sensory–tactile threshold could explain the tactile defensiveness usually seen among children with ASD [30].

Interestingly, migraineurs and people with ASD seem to have both an increased pain sensitivity and an increased sensitivity to other sensory stimuli [31].

However, the clinical experiments about pain sensitivity in ASD mainly investigate the tactile thermal and pain sensitivity, while migraine headache has a different, partly clarified, complex pathophysiology leading to the activation of meningeal nociceptors [32]. Unfortunately, there is a lack of literature evidence about the differences in pathogenesis and perception of migrainous pain in people with ASD compared to healthy individuals.

A further crucial element to be considered is that pain in humans has an undeniable emotional and social dimension. Thus, the expression of pain includes several manifestations such as facial expression changes, verbal activity, posture, movement and behavior. It is therefore reasonable to expect differences in reports of pain by subjects with ASD.

Nevertheless, recent studies have examined the pain reactions in children with ASD finding no significant difference in the number of facial, behavioral and physiological reactions between children with ASD and typically developing children [33,34].

In order to overcome the possible communication barriers, Failla et al. investigated the neural responses to pain in individuals with ASD using functional magnetic resonance imaging. Similarly to the aforementioned evidence, they found no statistically significant difference in pain ratings and neural pain signature responses during acute pain, while they observed a reduction in neural pain signature responses during sustained pain and after stimulus offset [35].

## 5. Excess of Endogenous Opioids Theory

The opioid system plays a crucial role in the nociception, in the analgesia and in the response to stress, as well as to affective processing, pleasure, reward, mood and the sense of well-being [36].

Some authors have suggested an excess of endogenous opioids to sustain the alleged reduced sensitivity to pain [37].

An exogenous opioid supply is able to determine behavioral effects, such as insensitivity to pain, stereotypical behavior, affective lability and reduced socialization [38,39].

However, subsequent studies failed to replicate the findings of opioid peptide excess putting this theory in doubt [40]. Moreover, some authors rather than individuating an excess of endogenous opioid neuropeptides related the reduced sensitivity to pain to different physiological and biological stress responses of patients with ASD [41]. Nevertheless, a recent systematic review reaffirmed that there was not sufficient evidence that an endogenous opioid imbalance has an impact on the core symptoms of autism in the majority of cases, although an exogenous administration of opioid antagonists can improve hyperactivity and restlessness in some subgroups of ASD children [42].

Most opioid analgesics are used as a second or third tier treatment for migraine. Most common opioid analgesics target the μ-opioid receptor, but μ-opioid agonists usually have low efficacy and they can contribute to the progression of migraine to a chronic and refractory condition [43].

However, some preclinical studies suggest that other members of the opioid receptor family can be better alternatives to μ-based approaches.

δ-receptors are mainly expressed in several regions involved in headache, including trigeminal and dorsal root ganglia, trigeminal nucleus caudalis, cortex, hippocampus, hypothalamus and amygdala. Their expression in limbic regions supports their role in emotional regulation [44].

δ agonists have fewer rewarding behaviors and less physical dependence, respiratory depression and constipation compared to μ-agonists. Moreover, literature evidence showed promising effects in multiple chronic pain models, including nitroglycerin evoked hyperalgesia, conditioned place aversion and cortical spreading depression [45].

Moreover, the positive effects of δ-agonists on emotional modulation may be beneficial in migraineurs considering the high comorbidity between headache and emotional disorders [46].

The κ-receptors are highly expressed in regions related to mood, motivation and pain such as the cerebral cortex, hippocampus, hypothalamus, nucleus accumbens, periaqueductal grey, spinal cord and dorsal root ganglia [47].

κ-receptor inhibitors emerged as an effective and well-tolerated therapeutic option for headache, probably limiting the κ-opioid peptide dynorphin recognized as a marker of stress that is considered the most common migraine trigger [48]. 

## 6. Serotonin Theory

Militerni el al. proposed an alternative theory to explain the reduced pain reactivity in individuals with ASD. They found a significant reduction in serotonemia in children with ASD and an abnormal pain reactivity [24]. Both autism and migraine are characterized by serotonergic abnormalities and by a hyperexcitable cortex [49]. However, ASD is characterized by insufficient or excessive serotonin signaling, suggesting a bidirectional serotonin involvement [50].

There is evidence that children with ASD have a lower initial capacity to produce serotonin in the central nervous system but they maintain a constant level of production, whereas this capacity declines with age in children with a normal neurodevelopment. Therefore, individuals with autism develop hyperserotonemia [51].

Serotonin plays a crucial role in promoting synaptogenesis and the formation of dendritic spines in cortical and striatal neurons [52]. Both high and low anormal serotonin levels in the brain during corticogenesis have been demonstrated to cause a disruption of synaptic connectivity, a potential neurobiological mechanism of ASD [53,54].

Serotonin is also a crucial neurotransmitter in the etiopathogenesis of migraine so that it is classically considered a “low serotonin syndrome”. In fact, patients with migraine have a reduction in levels of serotonin and tryptophan (precursor of serotonin) [55]. Most of serotonin is produced in the periphery, especially in the gastrointestinal system by enterochromaffin cells that release it in the bloodstream and its serum levels are regulated by the uptake into platelets [56]. The chronic peripheral low serotonin level reflects dysfunctions not only in blood platelets, but primarily in the brain [57]. The low serotonin state determines enhanced cortical spreading depression waves and increased neuronal activation within the trigeminal nucleus caudalis [58]. This mechanism is responsible for the cascade of events that ultimately leads to the activation of pain-sensitive trigeminovascular fibres probably thanks to an increased cortical excitability and sensitivity of the trigeminovascular pathway [59,60].

Therefore, a role of the disrupted serotonin system in the cortical control of nociceptive processing in patients with autism cannot be excluded. In fact, there is anecdotal evidence that sumatriptan (primarily a 5-HT1d receptor agonist used as an antimigraine medication) improves both symptoms of autism and migraine headaches when taken by patients suffering from both disorders [61].

Lastly, there are no reports about the effects on headache of selective serotonin reuptake inhibitors (SSRIs) primarily prescribed to treat depression, anxiety and obsessive-compulsive behaviors which are often comorbid with ASD [62].

Together with the dysfunction of serotonergic system-altered immune responses (see below) play a role in the pathogenesis of ASD. Both systemic and central proinflammatory immune stimuli contribute to the activation of the cerebral kynurenine pathway that was involved in several neurological disorders [63].

The kynurenine pathway is a metabolic pathway leading to the production of nicotinamide adenine dinucleotide and other active metabolites, from the degradation of more than 90% of the tryptophan metabolism.

The activation of the kynurenine pathway diverts tryptophan from the 5-HT synthesis route and depletes systemic tryptophan. Moreover, in people with ASD, kynurenic acid is significantly lower and kynurenine aminotransferase activity is decreased indicating high levels of neurotoxicity [64].

Nonetheless, some evidence suggests an interesting role of the kynurenine pathway in the pathogenesis of migraine. Kynurenic acid has a neuroprotective role blocking glutamate release, that is a neurotransmitter crucial in the migraine pathogenesis and plays a central role in the genesis of cortical spreading depression [65].

Moreover, kynurenines are important in the transmission of sensory impulses in the trigeminovascular system through NMDAR and AMPAR receptors. Additionally, there is evidence that kynurenine metabolites show significant serum reductions in patients with chronic migraine similarly to patients with ASD [66]. Nevertheless, adequate levels of kynurenic acid seem to reduce the sensitivity of the cerebral cortex to cortical spreading depression [67].

## 7. The Neuroinflammation Theory

Another dysregulated system in common in people with migraine and in people with autism is the immune response.

Increasing evidence shows that neuroinflammation is involved in the etiopathogenesis of neuropsychiatric disorders [68]. In autism an increased number of reactive microglia and astrocytes has been reported in postmortem tissues and in animal models [69]. Numerous studies investigate the immune-mediated response in autism often with contrasting results. However, in autism, a chronic inflammation seems to be in accordance with to the cytokine profile including both inflammatory and anti-inflammatory agents. A recent systematic review and meta-analysis, aiming to investigate the characteristics of the abnormal cytokine profile, found the following anomalies: interleukin (IL)-1beta (*p* = 0.001), IL-6 (*p* = 0.03), IL-8 (*p* = 0.04), interferon-gamma (*p* = 0.02), eotaxin (*p* = 0.01) and monocyte chemotactic protein-1 (*p* = 0.05) were significantly higher in patients with ASD, while concentrations of transforming growth factor-β1 were significantly lower (*p* = 0.001) [70].

Brain inflammation is also strongly linked to several pain disorders [71].

There is poor modern evidence to suggest a role of the neuronal inflammatory response in conjunction with acute migraine attacks. It would seem that repeated episodes of cortical spreading depression lead to a brain inflammation response [72]. This response is called by some authors “neurogenic neuroinflammation”, which is a sterile inflammation caused by a continuous stimulation of C and Aδ fibres leading to the release of neural mediators, mainly calcitonin gene-related peptide (CGrP) but also substance P provoking a trigeminal sensitization [73]. This cascade leads to the activation of glial cells and the alteration of several cytokine levels such as tumor necrosis factor (TNF), IL-1β and IL-6 which have been linked to migraine pathophysiology [74], similarly to the alterations found in patients with ASD.

Theoharides et al. proposed a link between autism and migraine in the involvement of mast cells. Corticotropin-releasing factor and neurotensin are significantly increased in the serum of ASD children and stimulate mast cells located perivascularly close to neurons and microglia to produce neurotoxic mediators [75]. Mast cells have also been implicated in the pathogenesis of migraine by participating in neurogenic inflammation [76]. Mast cells could act as a “mirror” of heterogeneous triggers stimulating microglia and they together secrete inflammatory molecules contributing to the risk of developing ASD and migraine [75].

## 8. Autism as a Minicolumnopathy 

Numerous studies have clarified that the pathogenic mechanisms behind ASD have a wide and heterogeneous genetic component [77]. This genetic vulnerability causes different and nonspecific neuroanatomical abnormalities such as disorganized gray and white matter, abnormal neuronal connectivity, regional anomalies of brain structure, differences in the number and volume of neurons, increased neuropil, vascular and glial abnormalities [78].

However, these patterns of atypical brain architecture show marked heterogeneity across individuals with ASD.

Some authors suggest that certain migratory and proliferative defects are at the root of etiopathology of autism. They consider ASD a minicolumnopathy. A minicolumn is a structural elementary unit of the neocortex composed principally by pyramidal cells with its own ecosystem of afferent, efferent and interneuronal connections [79].

Minicolumns in autism are abnormal, they are smaller and increased in total numbers and they present an irregular peripheral neuropil space. The peripheral neuropil space surrounds the minicolumn core and contains GABAergic interneurons, which are thought to protect the minicolumn by excessive excitatory inputs of neighboring minicolumns with repercussions on excitatory/inhibitory balance. This mechanism may in part explain the decreased seizure threshold seen in patients with autism [80,81]. A reduction in GABAergic inhibitory activity would be responsible for the high incidence of seizures and for hypersensitivity which is typical of patients with autism [82].

According to Casanova et al., minicolumnopathy in autism, the resultant excitatory/inhibitory imbalance and serotonergic abnormalities are the common keys to explain the autism core symptoms and gastrointestinal symptoms frequent in patients with ASD. In the light of this theory, the gastrointestinal phenomenology in patients with ASD is nothing more than a migraine equivalent—i.e., an abdominal migraine [83]. An abdominal migraine is a frequent cause of chronic and recurrent abdominal pain in children and it is often accompanied by other symptoms such as headache, pallor, anorexia, nausea, vomiting, photophobia, partial bowel obstruction and irritable bowel syndrome [84].

Therefore, this theory is supported by a common wide clinical phenomenology between migraineurs and patients with autism especially gastrointestinal complaints (see below).

## 9. Dysfunctional Gut–Brain Axis

Although headache and local pain do not represent frequent causes of complaints in children with ASD, there is a wide range of gastrointestinal signs and symptoms present both in migraine and autism.

Gastrointestinal problems are the most frequent medical condition associated with autism [85] and they are correlated with autism severity, worse behaviors and they can trigger regression in children with ASD [86]. The most frequent symptoms are alterations in bowel habits, mainly constipation, recurrent abdominal pains, bloating, nausea or vomiting, reflux and diarrhea. Moreover, these recurrent attacks often occur in a setting of phonophobia/photophobia and they are associated with autonomic symptoms such as pallor and flushing [87].

The causes of this gastrointestinal symptomatology are to be found in a dysfunctional gut–brain axis.

There is much literature evidence that children with autism have an altered gut microbiota, particularly in the relative amount of bacterial phyla [88]. The dysbiosis is associated in patients with ASD with an alteration of the intestinal mucosa barrier with a consequent increased permeability to exogenous molecules [89]. These molecules pass into the blood stream and they could be able to provoke a system immune response that alters the blood–brain barrier causing neuroinflammation with excessive microglial activation and increased proinflammatory cytokines [90].

Moreover, some bacterial species (*Lactobacillus*, *Streptococcus*, and *Lactococcus*) are able to produce serotonin and short-chain fatty acids altering the serotoninergic balance which is associated with gastrointestinal symptoms and this neurotransmitter plays a significant role in the etiology of migraine and autism representing a link in the brain–gut microbiome axis in ASD pathophysiology [91] (see also above).

Evidence also suggests complex and not entirely clear relationships between migraine and the gut–brain axis. Several factors influence both the gut–axis and central nervous system as cytokines (IL-1β, IL-6, IL-8, and TNF-α), gut microbiota, neuropeptides as serotonin, CGRP, substance P, vasoactive intestinal peptide, and neuropeptide Y that can have an antimicrobial impact on the gut microbiome [92].

Moreover, there is a high comorbidity between chronic migraine and gastrointestinal disorders such as helicobacter pylori infection, irritable bowel syndrome, inflammatory bowel disease and celiac disease, and there is evidence that in some cases the eradication of helicobacter pylori or a gluten-free diet or other diet strategies (probiotics, omega-3, fibers, vitamin D) may have an impact on the migraine course [93,94,95,96]. 

## 10. Genetic Susceptibility

Migraine and autism partly share a common genetic load provoking the involvement of neural networks which are coming sharply into focus.

A recent study by Sener et al. has investigated the expression of pain candidate genes in children with ASD. They found alterations in the mRNA in the peripheral blood of patients with ASD in the expression of HTR1E, OPRL1, OPRM1, TACR1, PRKG1, SCN9A and DRD4 genes supporting the relationship between an altered sensitivity to pain and autism [97].

Neurodevelopmental disorders have, almost invariably, a complex polygenic susceptibility that sometimes involves shared heritable factors with migraine. 

The strong genetic evidence about migraine disorders regards hemiplegic migraine, which is a rare monogenic subtype characterized by migraine with aura associated with hemiparesis mainly due to mutations in three genes—CACNA1A, ATP1A2 and SCN1A (other pathogenic variants are found in PRRT2, PNKD, SLC2A1, SLC1A3, SLC4A4 genes) [98].

Interestingly, there is literature evidence that all the three abovementioned genes or their homologs are involved in familial or sporadic forms of autism. However, no headache symptomatology was reported in association with autism for SCN1A [99,100] or ATP1A2 [101] genes, although we do not know if it was investigated. Only for CACNA1A and their homolog genes (i.e., CACNA1C gene responsible for Timothy syndrome) was the coexistence between autism and migraine reported [102,103].

Familial hemiplegic migraine mutations in SCN1A are commonly missense and cause gain-of-function effects such as a retarded inactivation, increased threshold-near persistent current, faster recovery and higher availability during repetitive stimulation [104].

On the contrary, heterozygous loss-of-function mutations in the SCN1A gene provoke severe myoclonic epilepsy in infancy in which patients have autism-spectrum behaviors [105].

Various ion channel gene defects altering normal ion flux across the neuronal membrane cause harmful effects to the generation of action potentials, abnormalities in early brain development, in gene expression and in cell morphology [106].

Channelopathies, the diseases caused by the disturbed function of ion channel subunits, have become, supported by growing literature evidence, significantly more relevant in the pathogenesis of neurodevelopmental disorders. Calcium signaling plays a crucial role in the pathogenesis of neuropsychiatric diseases, including migraine headache, cerebellar ataxia, autism, schizophrenia, bipolar disorder and depression [107], but sodium, potassium and chloride channels are also implicated [99,108]. Channelopathies result in an altered excitation/inhibition balance that is responsible for the brain dysmaturation typical of neurodevelopmental disorders [109,110]. This imbalance determines a neuronal hyperexcitability leading to seizures and epilepsy that are highly comorbid with ASD and make the brain more susceptible to cortical spreading depression, the pathophysiological mechanism at the base of aura symptoms [111].

For instance, most missense variants of calcium channels have gain-of-function effects, provoking increased Ca^2+^ influx and determining enhanced glutamatergic neurotransmission, neuronal hyperexcitability and increased susceptibility to cortical spreading depression [112].

Mutations of calcium channels and the dysregulation of their functions are also able to impair the synaptic plasticity and the regulation of synaptic strength in a wide range of neuropsychiatric diseases including ASD [107].

## 11. Conclusions

Even if autism and migraine are two common neurological conditions, only a few studies investigate their comorbidity. These studies, despite a small sample of patients with autism, indicate a high rate of migrainous symptomatology. Individuals with autism frequently have an altered pain sensitivity that could distort their perception of headaches. Moreover, the social dimension of pain could be impaired in people with autism with unforeseeable consequences in reports of pain.

Autism and migraine share common pathophysiological changes: neurotransmission dysregulation, especially of the serotoninergic system; altered immune response causing neurogenic neuroinflammation; abnormal findings especially in the cortical minicolumn organization and in the dysfunctional gut–brain axis; shared susceptibility genes.

Regarding the comorbidity between autism and migraine, it seems clear that further epidemiological studies are needed to take into account the true scale of this poorly explored association.

## Figures and Tables

**Table 1 brainsci-10-00615-t001:** Autism spectrum disorder (ASD)-migraine comorbidity studies.

	ASD Sample	Rate of Migraine	Other Findings
**Underwood et al., 2019** [21]	105 adults (76 healthy controls) with no intellectual disability	42.7% (vs. 20.5% of controls)	High rate (89.5%) of psychiatric comorbidities (depression 62.9%; anxiety 55.2%)
**Victorio, 2014** [22]	18 children	61%	44% without aura; 5,6% with aura and 11% both types
**Sullivan et al., 2014** [23]	81 children	28.4%	more generalized anxiety and sensory hyperreactivity

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
