# Peer review of "Autism and Migraine: An Unexplored Association?"

_brainsci, 2020, doi:10.3390/brainsci10090615_

Round 1
Reviewer 1 Report
The author provides a very interesting and well-written study investigating the connection between migraine and autism.
I don't have to suggest any changes with respect to the content of the paper.
I just have some minor comments with respect to the format of the paper.
Please check once more the punctuation rules for English and review the whole text. There is an excessive use of commas that needs to be reviewed and corrected.
Please check carefully the reference list and edit the spacing in all of the references provided in order to be consistent. Please indicate the complete name of each Journal. Please notice that before the year of publication.
Author Response
Dear reviewers,
I would like to thank the reviewers for your valued comments and suggestions to the article. As you requested, we made all the necessary changes in our manuscript to address the reviewers’ concerns and we detailed below how the points raised by the referees have been accommodated. The main changes are written in red in the text of the manuscript. From the changes made in the revised manuscript and responses provided below, I hope you are convinced that we have adequately addressed the reviewer’s concerns and made the paper better. If there are any further questions, please feel free to let me know.
Reviewer 1
The author provides a very interesting and well-written study investigating the connection between migraine and autism.
I don't have to suggest any changes with respect to the content of the paper.
I just have some minor comments with respect to the format of the paper.
Please check once more the punctuation rules for English and review the whole text. There is an excessive use of commas that needs to be reviewed and corrected.
Thanks for the suggestion. The punctuation of the whole text has been revised and corrected.
Please check carefully the reference list and edit the spacing in all of the references provided in order to be consistent. Please indicate the complete name of each Journal. Please notice that before the year of publication
Thanks for the suggestion. All references have been reformatted using the bibliography software EndNote and the MDPI style file (https://endnote.com/style_download/mdpi/https://endnote.com/style_download/mdpi/).
Moreover, please note that according to “Instructions for Authors” (https://www.mdpi.com/journal/brainsci/instructions) the name of journal should be abbreviated:
- Author 1, A.B.; Author 2, C.D. Title of the article. Abbreviated Journal NameYear, Volume, page range.
Reviewer 2 Report
Luigi Vetri explores the association between autism and migraine in this narrative review. Dr. Vetri summarizes the major studies that examine the co-occurrence of ASD and migraine in adult and pediatric patients. The review then explores a number of theories regarding the etiopathology of autism and seeks to link these theories with migraine pathology. Overall, this is an intriguing topic that deserves more attention, and this is a reasonable review, although this manuscript could be greatly improved in some areas with a more comprehensive and nuanced discussion.
Major points:
- Introduction: The most compelling reason to explore the association between migraine and autism is not that they are common diseases. Rather, both diseases have a central feature of sensory processing abnormalities. This should be discussed in the introduction.
- Section 4.2, "Excess of endogenous opioids theory" There is no discussion of the role of opioids in migraine. The author could consider discussing the potential role of delta and kappa opioid receptors in migraine treatment.
- Section 4.3 "Serotonin theory". I recommend including a discussion on disruption of the kyurenine pathway in autism and migraine
- Section 6 "Dysfunctional gut-brain axis" There is no discussion of migraine in this section
- Section 7 "Genetic susceptibility" SCN1A- please add to the discussion that mutations in this sodium channel are likely gain of function in migraine vs. likely loss of function in autism; re: Calcium signaling- discuss how mutations in calcium channels involved in autism and migraine are likely gain of function mutations/polymorphisms that potentiate synaptic signaling: thus disrupting synaptic plasticity in autism and lowering CSD threshold in migraine; Consider adding a discussion on the role of connexin in astrocytes that may be implicated in migraine and autism.
Minor:
Subsection "4." is missing a title, What is the difference between subsections in 4 vs. Subsection 5,6,7?
Author Response
Dear reviewers,
I would like to thank the reviewers for your valued comments and suggestions to the article. As you requested, we made all the necessary changes in our manuscript to address the reviewers’ concerns and we detailed below how the points raised by the referees have been accommodated. The main changes are written in red in the text of the manuscript. From the changes made in the revised manuscript and responses provided below, I hope you are convinced that we have adequately addressed the reviewer’s concerns and made the paper better. If there are any further questions, please feel free to let me know.
Reviewer
Luigi Vetri explores the association between autism and migraine in this narrative review. Dr. Vetri summarizes the major studies that examine the co-occurrence of ASD and migraine in adult and pediatric patients. The review then explores a number of theories regarding the etiopathology of autism and seeks to link these theories with migraine pathology. Overall, this is an intriguing topic that deserves more attention, and this is a reasonable review, although this manuscript could be greatly improved in some areas with a more comprehensive and nuanced discussion.
I appreciated your suggestions because they are highly constructive and fruitful.
Major points:
Introduction: The most compelling reason to explore the association between migraine and autism is not that they are common diseases. Rather, both diseases have a central feature of sensory processing abnormalities. This should be discussed in the introduction.
Thanks for the suggestion. I added to introduction a part about sensory processing abnormalities in both conditions (you can see it in lines 92-113).
Section 4.2, "Excess of endogenous opioids theory" There is no discussion of the role of opioids in migraine. The author could consider discussing the potential role of delta and kappa opioid receptors in migraine treatment.
Thanks for the suggestion. I better expanded the role of opioids in migraine and I mentioned the delta and kappa opioid receptors as possible new target in the migraine treatment (please see the lines 199-218).
Section 4.3 "Serotonin theory". I recommend including a discussion on disruption of the kyurenine pathway in autism and migraine
I agree with your suggestion. I discussed about kyurenine pathway in autism and migraine in lines 251-270.
Section 6 "Dysfunctional gut-brain axis" There is no discussion of migraine in this section
I expanded the discussion about the relationship between gut-brain axis and migraine (please see the lines 353-362)
Section 7 "Genetic susceptibility" SCN1A- please add to the discussion that mutations in this sodium channel are likely gain of function in migraine vs. likely loss of function in autism; re: Calcium signaling- discuss how mutations in calcium channels involved in autism and migraine are likely gain of function mutations/polymorphisms that potentiate synaptic signaling: thus disrupting synaptic plasticity in autism and lowering CSD threshold in migraine; Consider adding a discussion on the role of connexin in astrocytes that may be implicated in migraine and autism.
Thanks for your suggestions. I discussed the role of different mutations in sodium and calcium channels in lines 382-386 and 400-405. Regarding the connexins, in my opinion, future studies are needed to clarify the exact role of these proteins in the pathogenesis of headache.
Minor:
Subsection "4." is missing a title, What is the difference between subsections in 4 vs. Subsection 5,6,7?
I agree with your suggestion. All the subsections now have a title and they have been renumbered.
Reviewer 3 Report
In this review article, the author describes a link between autism spectrum disorder and migraine. He makes the point that there is only limited information available, including 3 studies directly assessing migraine symptomatology in autism patients (both adults and children).
Although investigation of the connection between autism disorders and migraine is crucial both for understanding the underlying mechanisms inducing these disorders but also advancement of treatment strategies, in my opinion the review is not yet in a shape that would allow for publication in its current form. Many parts are mentioned in a very superficial way, without going into detail and without really bridging the two topics. Other parts are mere listing of facts and do not provide further information. In addition, a large part of the review focusses on altered pain perception in autism spectrum disorders, which of course is relevant as there exists a strong link between general pain and migraine disorders, but like this only represents a lose indirect link.
I would recommend reworking of the manuscript, first better describing the definitions and descriptions of both disease types in more depth, and then emphasising the overlap between both disorders side by side.
Author Response
Dear reviewers,
I would like to thank the reviewers for your valued comments and suggestions to the article. As you requested, we made all the necessary changes in our manuscript to address the reviewers’ concerns and we detailed below how the points raised by the referees have been accommodated. The main changes are written in red in the text of the manuscript. From the changes made in the revised manuscript and responses provided below, I hope you are convinced that we have adequately addressed the reviewer’s concerns and made the paper better. If there are any further questions, please feel free to let me know.
Reviewer
In this review article, the author describes a link between autism spectrum disorder and migraine. He makes the point that there is only limited information available, including 3 studies directly assessing migraine symptomatology in autism patients (both adults and children).
Although investigation of the connection between autism disorders and migraine is crucial both for understanding the underlying mechanisms inducing these disorders but also advancement of treatment strategies, in my opinion the review is not yet in a shape that would allow for publication in its current form. Many parts are mentioned in a very superficial way, without going into detail and without really bridging the two topics. Other parts are mere listing of facts and do not provide further information. In addition, a large part of the review focusses on altered pain perception in autism spectrum disorders, which of course is relevant as there exists a strong link between general pain and migraine disorders, but like this only represents a lose indirect link.
I would recommend reworking of the manuscript, first better describing the definitions and descriptions of both disease types in more depth, and then emphasising the overlap between both disorders side by side.
Thanks for your revision. According to your suggestions I discussed in more depth the common characteristics of migraine and autism:
- I added to the introduction a part about sensory processing abnormalities in both conditions (you can see it in lines 92-113).
- I expanded the discussion about the relationship between gut-brain axis and migraine (please see the lines 353-362)
- I better expanded the role of opioids in migraine and I mentioned the delta and kappa opioid receptors as a possible new target in migraine treatment (please see the lines 199-218).
- I discussed about the kyurenine pathway both in autism and migraine in lines 251-270
- I discussed the role of different mutations in sodium and calcium channels in autism and migraine in lines 382-386 and 400-405.
I hope that you will appreciate my efforts to make the paper better.
Round 2
Reviewer 2 Report
Content is greatly improved after revisions. Would like to see more discussion on future areas where the connection between autism and migraine might be strengthened, but that might be beyond the scope of the review.
Reviewer 3 Report
Although the author has added additonal paragraphs on different topics, i still think that the review in this form is not suitable for publication. Also taking the junior status of the author in general and especially in the field into account (8 pubmed indexed publications, only one as a leading author, only one review on autisms as a coauthor), i think the publication could strongly benefit from an experienced senior author overseeing the work and content of the manuscript.